# The ARTEMIS Center: An Environmental Health Prevention Platform Dedicated to Reproduction

**DOI:** 10.3390/ijerph17030694

**Published:** 2020-01-21

**Authors:** Fleur Delva, Guyguy Manangama, Patrick Brochard, Raphaëlle Teysseire, Loïc Sentilhes

**Affiliations:** 1Bordeaux Population Health Research Center, Inserm UMR1219-EPICENE, University of Bordeaux, 33076 Bordeaux, France; 2Environmental Health Platform Dedicated to Reproduction, ARTEMIS Center, 33076 Bordeaux, France; 3Department of Obstetrics and Gynecology, Bordeaux University Hospital, 33076 Bordeaux, France

**Keywords:** prevention, environmental exposure, reproduction

## Abstract

In France, a new approach is being developed through the ARTEMIS Center, which is a prevention platform for environmental health dedicated to reproduction. The objective is to describe the clinical management of patients in the ARTEMIS center. Couples with a condition affecting reproduction are referred to the ARTEMIS center. Management includes a medical consultation and a standardized interview. Current exposure is assessed by a questionnaire that includes exposure circumstances to reproductive risk factor and on the basis of which it is possible to implement preventive action in clinical practice without prejudging the role of such exposure in the onset of disease. From 16 February 2016 to 2 May 2019, 779 patients were seen in the ARTEMIS center. On the day of the consultation, 88.3% men and 72.2% women were employed. Among employed men, 61.5% had at least one instance of occupational exposure to a reproductive risk factor, and among employed women, 57.8%. The main nonprofessional circumstances of exposure identified were proximity of the residence to an agricultural area (35.3%) and domestic pesticide exposure (79.7%). The preventive actions implemented by the ARTEMIS center are targeted to the individual practices of patients. However, patient care also allows their physicians to become familiarized with environmental health.

## 1. Introduction

Environmental exposure is known to interfere with reproduction (fertility disorders, pregnancy pathologies, and developmental abnormalities) [1,2]. Exposure to certain chemical or physical agents (solvents, heavy metals, pesticides, drugs, ionizing radiation, heat) is associated with alterations of spermatic parameters [3,4,5], negative effects on fertility in women [6], spontaneous abortions, and congenital malformations [7,8], as well as certain adverse pregnancy outcomes (fetal death, intrauterine growth retardation (IUGR), and prematurity) [7]. In addition, environmental exposure during fetal life may increase the risk of chronic disease in childhood and adulthood (hypothesis of the developmental origins of health and disease (DOHaD)) [9]. Finally, exposure to environmental reproductive risk factors has been reported in the scientific literature to lead to transgenerational effects [10,11].

In this context, The American College of Obstetricians and Gynecologists and the American Society for Reproductive Medicine published a committee opinion calling “to identify and reduce exposure to toxic environmental agents while addressing the consequences of such exposure” [1]. The implementation of pre-conception visits, which include the evaluation of alcohol and smoking, is/has been recommended [12]. These visits would enable the evaluation of professional and extraprofessional exposure [12]. However, there are two major difficulties in the clinical management of environmental risk factors. (**1**) There are a very high number of chemical substances to which humans are exposed, and knowledge is still lacking for many. Certain substances are well identified (dangerous chemical agents classified as certainly or probably reproductive toxicants by the European Union [13]), but the reprotoxicity of others is still hypothetical (for example, certain substances among pesticides and organic solvents) [7]. A significant volume of research has been published in this area in recent years. (**2**) In clinical practice, physicians do not have the time to search for such exposure. In one study, less than 20% of obstetricians and interns reported routinely asking pregnant women about environmental exposure [14]. In addition, little information is provided to pregnant women about environmental exposure [15,16]. Indeed, the analysis of environmental exposure is a long, complex, and multidisciplinary task, and physician training in environmental medicine is limited. In France, a new approach is being developed through the ARTEMIS (Aquitaine, Reproduction, Environnement, Maternité, Enfance et Impact en Santé) Center, which is a preventive environmental health platform targeted to patients with conditions affecting reproduction. The objectives of the ARTEMIS Center are (**i**) to evaluate the environmental exposure (occupational and extraprofessional) of couples with conditions affecting reproduction: fertility disorders, pregnancy complications, or parents of children born with congenital malformations, (**ii**) propose preventive measures to reduce exposure to these risk factors, and (**iii**) educate health professionals about environmental health. Here, we describe the clinical management of such patients.

## 2. Methods

### 2.1. Prior Listing of Reproductive Risk Factors

Prior to implementing patient management in the ARTEMIS Center, a study to identify and prioritize chemical risk factors for human reproduction was carried out [17] (Figure 1). Briefly, reproductive risk factors were identified from relevant regulations or scientific reports or databases. The reproductive hazards were prioritized according to the strength of evidence concerning their impact on fertility or the development of offspring (Appendix A). The substances for which adverse health effects have been demonstrated in epidemiological studies or documented in robust animal studies were first prioritized. In parallel, a literature search was conducted to identify nonchemical risk factors. The approach resulted in the identification of twelve families of risk factors: physical constraints (prolonged sitting for men, carrying a load, prolonged standing, etc. for women), organizational constraints (shift work, night work), physical agents (ionizing radiation, a hot environment), pesticides, drugs, solvents, biological agents, complex fumes, polycyclic aromatic hydrocarbons (PAHs), chemical reagents, carbon monoxide, and metals. Then, each reproductive risk factor was linked to the relevant occupational and nonoccupational circumstances of exposure and a questionnaire developed that included all such exposure circumstances. The data collected in the questionnaire are medical history, lifestyle habits (alcohol/tobacco), professional routine and the risk factors associated with each job, and residential routine and nonoccupational exposure. This questionnaire is available on request by email (centre.artemis@chu-bordeaux.fr). However, it is currently only available in French.

### 2.2. Care Pathway

Couples with a condition affecting reproduction are referred to the ARTEMIS center by hospital practitioners, gynecologists, or pediatricians. Management includes a medical consultation and a standardized interview performed by a nurse or an engineer in environmental heath, based on a questionnaire that identifies potential activities that expose patients to reproductive risk factors. The consultation in which the standardized interview is conducted is not only a time for data collection but also for providing the first preventive advice.

An engineer in environmental heath then performs analysis in order to identify potential exposures to reproductive risk factors listed by the ARTEMIS Center in both environmental and occupational settings. The analysis is made on the basis on the information collected during the interview by the patients (describing mainly circumstances of potential exposure more than exposure itself, which is difficult to obtain from the subject). Analysis is eventually completed by other investigations such as the analysis of the safety data sheets of the products used, or in some cases, biometrology (measure of blood lead levels). No field observations are conducted. Exposures are described according to three criteria: the probability that this risk factor is actually present in the patient’s environment (>50% vs. ≤50%), the frequency at which it is found in the patient’s environment (>30% vs. ≤30%), and the intensity of risk factor exposure (high vs. low). It is difficult in clinical practice to identify nonoccupational exposure to chemicals, unlike that of occupational exposure. Indeed, this requires a lengthy and complex complementary investigation (exact references of the products used by the patients, requests for safety data sheets from the suppliers, analysis of the composition, etc.), which often results in obtaining only the approximate composition of the product. It is common in nonoccupational settings to identify only the circumstances of exposure that are documented in the scientific literature as potentially exposing individuals to one or more reproduction risk factors. Biological risks are investigated by clinicians in clinical practice, and thus the ARTEMIS center does not have an active approach to identify them. However, preventive advice is provided to women working in professions for which they are at risk for such exposure (e.g., nursery assistant).

After analysis by the engineer in environmental heath, medical conclusions with targeted preventive measures are proposed to each member of the couple after a multidisciplinary staff meeting with nurse, engineer in environmental heath, occupational physician, and public health physician. If occupational exposure is suspected, the consultation report is sent to the patient’s occupational health department, following the patient’s consent, for the implementation of preventive actions in the workplace. If the patient does not consent or his/her workplace does not have an occupational health service, the industrial hygienist conducts the hazard study and provides appropriate preventive advice. Preventive advice on appropriate actions is also provided to patients for nonoccupational exposure.

### 2.3. Population

Here, we describe all patients seen at the ARTEMIS Center between 16 February 2016 and 2 May 2019.

By convention, the term “patient” refers to
-A man, when consulting for male infertility-A woman, when consulting for female infertility, pregnancy complications, or congenital malformations.

Although the second member of the couple benefits from care at the ARTEMIS Center, the spouse’s data are not coded with the same precision in the patient file, and thus they are not presented herein.

### 2.4. Ethics

This paper is the description of the clinical management of patients. Therefore, it is the use of retrospective data collected for clinical management purposes. Patients are informed about the reuse of their data for research purposes through the hospital’s welcome booklet. They may object to it following this information.

### 2.5. Analyses

Quantitative analyses were carried out using SAS 9.4 software. Quantitative variables are described as means and standard deviations, and qualitative variables as numbers and frequencies. All analyses were conducted based on the available data and included the patients’ sociodemographic characteristics (age, education, and body mass index), diagnoses of pregnancy complications, lifestyle habits (alcohol/tobacco), and occupational and nonoccupational exposure, as well as the proposed preventive measures. Only exposures for which it is possible to implement preventive action in clinical practice without prejudging the role of such exposures in the onset of the disease are presented. Thus, all exposures to reproductive risk factors were investigated without trying to identify for a given risk factor whether it might have contributed to the onset of the pathology.

## 3. Results

From 16 February 2016 to 2 May 2019, 779 patients were seen in the ARTEMIS Center (Figure 2).

Among them, 676 (87%) were women. The average age of the men was 35.0 years, and that of the women 33.0 (Table 1).

Among the 302 couples who consulted for fertility disorders, 199 (65.9%) women and 103 (34.1%) men had a fertility disorder and were considered as patients. The principal diagnoses of pregnancy complications of women seen at the ARTEMIS center were complications associated with the fetus (270 (59.1%)) and congenital malformations (187 (40.9%)) (Table 2).

On the day of the consultation, 91 (88.3%) men were employed and 488 (72.2%) women. Among the men who were employed, 56 (61.5%) had at least one instance of occupational exposure to a reproductive risk factor, 28 (30.8%) had no occupational exposure to a reproductive risk factor, and it was not possible to determine exposure based solely on the data collected during the interview for 7 (7.7%). The most frequent occupational exposure was to pesticides (13, 14.3%), solvents (13, 14.3%), metals (8, 8.8%), polycyclic aromatic hydrocarbons (5, 5.5%), and physical agents (5, 5.4%) (Table 3). Among women, 57.8% (n = 282) had at least one instance of occupational exposure to a reproductive risk factor, 37.7 (n = 184) had no occupational exposure to a reproductive risk factor, and it was not possible to determine exposure based solely on the data collected during the interview for 4.5% (n = 22). The most frequent occupational exposure was to physical constraints (156, 32.0%), drugs (5.5%), organizational demands (23, 4.7%), and solvents (22, 4.5%) (Table 3). The sectors of activity and occupations of the exposed patients are presented in Table 4 and Table 5.

On the day of consultation, 34 (33.0%) men and 104 (15.4%) women were smokers and 26 (25.2%) men and 180 (26.6%) women were ex-smokers. On the day of consultation, 400 (69.4%) women and 83 (81.0%) men declared to consume alcohol occasionally or regularly. The principal nonprofessional circumstances of exposure identified were the residence in proximity to an agricultural area (268, 35.3%) and domestic pesticide exposure (605, 79.7%) (Table 6).

The sending of consultation reports to the occupational physicians was only registered from 15 April, 2017; 494 employed patients were seen between this date and 2 May, 2019. Among them, 289 (58.5%) were considered to have been occupationally exposed. In total, 139 (48.1%) letters were sent to occupational health services during this period, 53 (18.3%) patients did not agree to have their occupational health service contacted, and the employer of 40 (13.8%) did not have an occupational health service (data was missing for 57 patients). Other preventive measures implemented following the identification of occupational exposure or extraprofessional exposure are presented in Table 7. Concerning extraprofessional exposure, preventive advice was provided to the patients during the interview and in the report targeted to their activities. Smoking cessation is systematically proposed to the patients who smoke and their spouses.

## 4. Discussions

We identified occupational exposure to reproductive risk factors for nearly half of employed patients during the clinical management of ARTEMIS patients. We also found at least one circumstance of exposure to reproductive risk factors in the extraprofessional environment of patients. The main objective of the ARTEMIS Center is to put in place preventive action for the future and not to find a cause to explain the condition. Ways to change patient behavior are explained and targeted, based on the activities described by the patients, in a face-to-face interview, which can facilitate adherence. The objective is to influence the child’s development during the next pregnancy, as well as in the longer term, by reducing exposure to reproductive risk factors (physical and chemical). The hypothesis of the developmental origins of health and disease implies that fetal exposure to chemicals affects health and chronic diseases throughout their lives [18,19]. Thus, in 2015, International Federation of Gynecology and Obstetrics (FIGO) recommended implementing measures to limit preconceptional and prenatal exposure to toxic chemicals [1]. However, at the ARTEMIS Center, we only work on changing individual behavior. Limiting exposure to environmental factors should not only be conducted at the individual level. Other actions must also be implemented through (**1**) regulation, which makes it possible to avoid widening social inequalities in health by not stigmatizing people who are not in a behavioral change process; (**2**) communication on the appropriate actions to adopt, issued directly to the general population (for example, advice to young adults of childbearing age on the abandonment of unfavorable practices, such as the use of indoor pesticides, candles, and air fresheners); and (**3**) a specific approach to risk prevention at work, in particular by modifying or adapting the jobs of women who wish to conceive [20]. Indeed, more than 50% of the patients received at the ARTEMIS Center were shown to have been exposed to reproductive risk factors in the workplace. In France, there are regulatory provisions to protect pregnant women, but reproductive risk is poorly known and, therefore, poorly identified, both by occupational health services and employers and their employees [20]. As a result, the preventive and protective measures associated with such risk are often inadequate or nonexistent (especially for men). The prevention of reproductive risk in the workplace is almost always associated with pregnancy, often by limiting exposure to carcinogens, but not through actions based on the reproductive risk factors themselves [21]. Thus, reproductive health prevention remains limited to the second and third trimester of pregnancy alone and ignores the critical preconceptual and periconceptual phases. However, the value of implementing preventive measures has been demonstrated in Canada, with a reduction in the risk of preterm birth and low weight for gestational age for women exposed to physical and organizational constraints in the workplace [22,23]. Exposure in nonprofessional settings is generally low but comprises exposures to multiple substances that are difficult to identify in daily clinical practice. Indeed, there is biological plausibility of a health effect related to such exposure. Experimental data show that exposure to chemicals at low levels can lead to endocrine disruption [2,10,24]. The objective of the ARTEMIS Center is to provide information that enables patients to adopt the appropriate behaviors for better health. As a result, health education, as defined by the WHO, has been implemented [25].

Alcohol consumption and tobacco smoke remain the main risk factors for reproduction [26,27,28]. Evidence in the scientific literature is beginning to show possible interactions between tobacco exposure and environmental exposures and even with exposures to passive maternal smoking [29]. It is necessary to take them into account in the evaluation of patients’ environmental exposures, especially for a platform dedicated to reproduction. The use of a method that has proven its effectiveness in initiating smoking cessation helps the investigator during the interview. In addition, it is important to prioritize with the patient the levels of risk according to the exposures and the level of exposure to the different risk factors on reproduction. Nutrition is also an important risk factor to consider [30]. It is not yet investigated in the ARTEMIS center as it is managed directly by the clinicians.

The creation of the ARTEMIS Center is an innovative action, for which an evaluation process was carried out in 2018. This center meets several objectives of the French National Health Strategy 2018–2022, in particular that of “making users aware of the behaviors to adopt to reduce their emissions and their exposure to environmental risk, particularly in the most exposed territories”. Four other platforms have since been set up in France following the same model as the ARTEMIS Center (at the Centre Hospitalier Intercommunal de Créteil, the Fernand Widal Hospital of the Assistance Publique-Hôpitaux de Paris, the CHU de Rennes, and the Assistance Publique-Hôpitaux de Marseille). This network constitutes the PREVENIR (Prevention, Environment, and Reproduction) platforms, with the common mission of integrating environmental health into the clinical management of patients in France. In addition, in France, the national strategy on endocrine disrupters 2 has just been published. One of its objectives is to train health professionals in environmental health so that they can inform their patients about this topic. Patient management through the PREVENIR platforms makes it possible to familiarize physicians with the field of environmental health.

## 5. Conclusions

ARTEMIS Center is an innovative action in prevention. The preventive actions implemented by the ARTEMIS center are targeted to the individual practices of patients. However, patient care also allows their physicians to become familiarized with environmental health.

## Figures and Tables

**Figure 1 ijerph-17-00694-f001:**
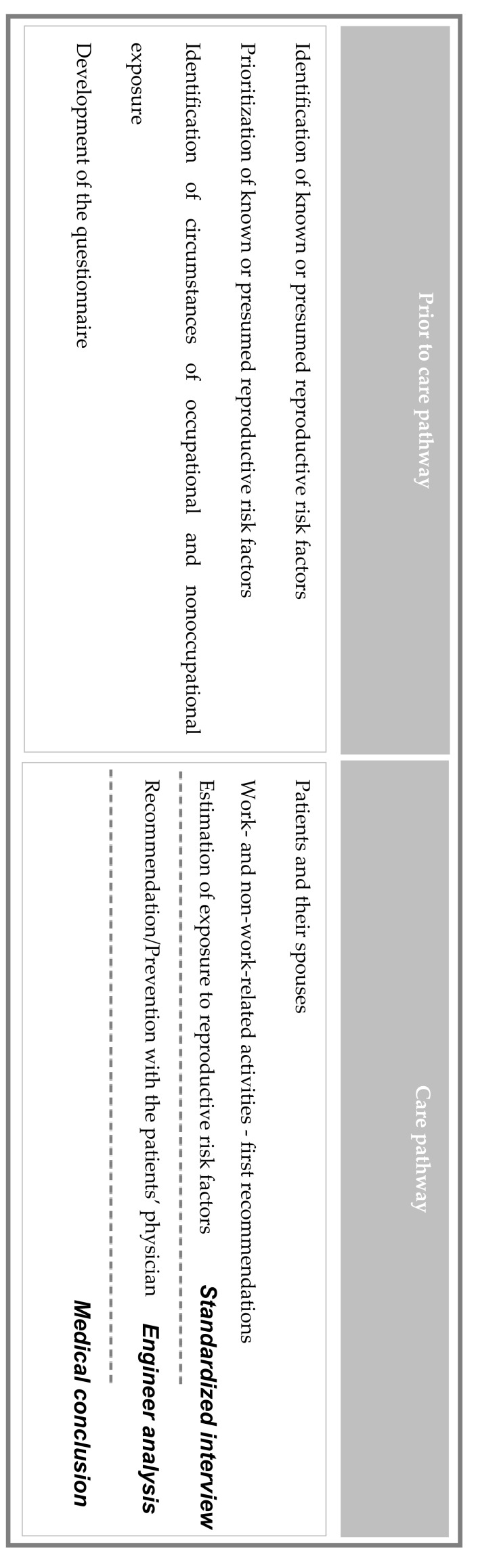
Measures implemented at the ARTEMIS Center.

**Figure 2 ijerph-17-00694-f002:**
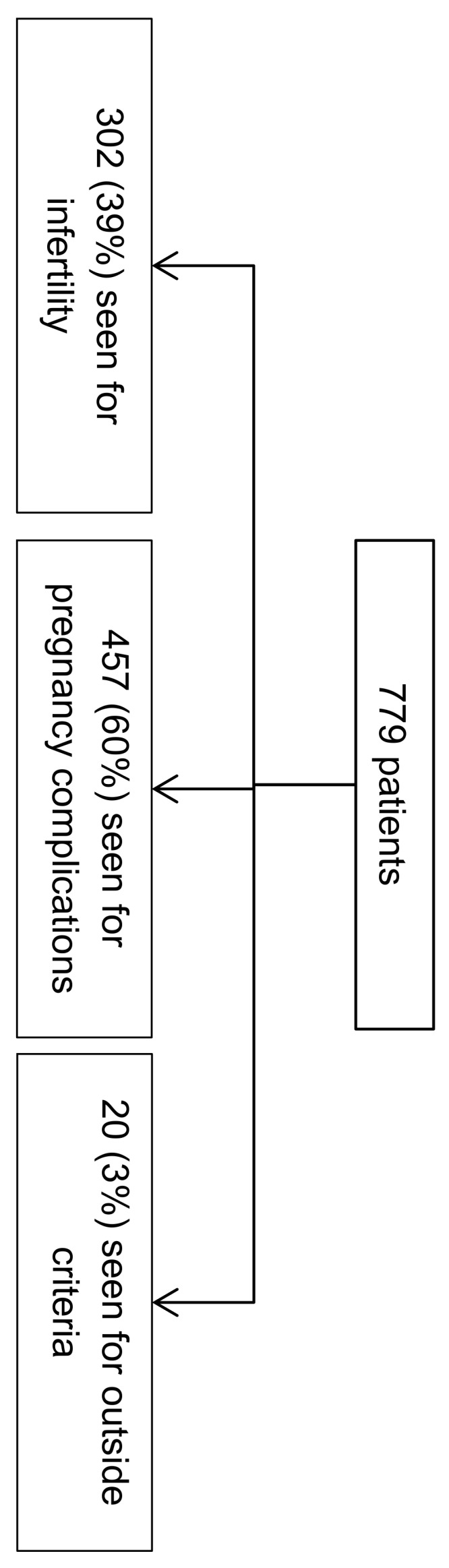
Patients were seen in the ARTEMIS center from 16 February 2016 to 2 May 2019.

**Table 1 ijerph-17-00694-t001:** Sociodemographic characteristics of the patients seen at the ARTEMIS Center from 16 February 2016 to 2 May 2019 (n = 779), according to gender.

	**Males (n = 103)**		**Women (n = 676)**	
**Age (Years) m, SD**	**35.0 ± 6.6**		**33.0 ± 5.0**	
	**n**	**%**	**n**	**%**
**BMI ***				
Underweight	0	0.0	31	6.6
Healthy weight	23	34.3	279	55.1
Overweight	29	43.3	104	20.6
Obese	15	22.4	92	8.2
**Education Level**				
Master’s or equivalent	22	21.4	116	30.4
Short Cycle Higher Education	22	21.4	133	34.8
High School	6	5.8	18	4.7
Professional High School	9	8.7	34	8.9
Professional qualification	30	29.1	54	14.1
Unknown	5	4.9	13	5.2
Without certificate	9	8.7	34	5.0

* missing data: males n = 36, women n = 170. BMI: body mass index.

**Table 2 ijerph-17-00694-t002:** Diagnosis of fertility disorders and pregnancy complications of patients seen at the ARTEMIS center from 16 February 2016 to 2 May 2019 (n = 457).

Diagnosis of Fertility Disorders and Pregnancy Complications	n	%
Oligoasthenoteroteratospermia	100	21.9
Ovarian insufficiency	64	14.0
Other fertility disorders	138	30.2
Pregnancy with abortive outcome	74	16.2
Hypertensive disorders and diabetes mellitus	107	23.4
Premature rupture of membranes. Placental disorders	100	21.9
False labor	33	7.2
Fetus complications	180	39.4
Congenital malformations of the circulatory system	76	16.6
Congenital malformations of the nervous system	5	1.1
Congenital malformations of eye, ear, face, and neck	4	0.9
Congenital malformations of the respiratory system	3	0.7
Cleft lip and cleft palate	6	1.3
Other congenital malformations of the digestive system	9	2.0
Congenital malformations of genital organs	16	3.5
Congenital malformations of the urinary system	20	4.4
Others malformations:		
Congenital malformations and deformations of the musculoskeletal system	12	2.6
Other congenital malformations	17	3.7
Chromosomal abnormalities, not elsewhere classified	2	0.4

**Table 3 ijerph-17-00694-t003:** Occupational exposure reported on the day of consultation for employed patients seen in interviews at the ARTEMIS center from 16 February 2016 to 2 May 2019 (n = 91 employed men and 488 employed women).

	Total		Probability				Frequency				Intensity			
			>50%		≤50%		>30%		≤30%		High		Low	
	n	%	n	%	n	%	n	%	n	%	n	%	n	%
**Men**														
Physical constraints	4	4.4	3	75.0	1	25.5	4	100.0	-	-	1	25.0	1	25.0
Physical agents	5	5.4	4	80.0	-	-	1	20.0	2	40.0	1	20.0	2	40.0
Pesticides	13	14.3	10	76.9	3	23.1	7	53.8	5	38.5	4	30.8	9	69.2
Drugs	1	1.1	-	-	1	100.0	-	-	1	100.0		-	1	100.0
Solvents	13	14.3	7	53.8	6	46.2	3	23.1	10	76.9	3	23.1	10	76.9
Biological agents	-	-	-	-	-	-	-	-	-	-	-	-	-	-
Complex fumes	1	1.1	1	100.0	-	-	1	100.0	-	-	1	100.0	-	-
Polycyclic aromatic hydrocarbon	5	5.5	5	100.0	-	-	3	60.0	1	20.0	1	20.0	3	60.0
Chemical reagents	-	-	-	-	-	-	-	-	-	-	-	-	-	-
Carbon monoxide	1	1.1	1	100.0	-	-	1	100.0	-	-	-	-	1	100.0
Metals	8	8.8	-	-	8	100.0	1	12.5	7	87.5	5	62.5	1	12.5
**Women**														
Physical constraints	156	32.0	143	91.7	13	8.3	93	59.6	63	40.4	65	41.7	21	13.5
Organizational constraints	23	4.7	19	82.6	4	17.4	13	56.5	8	34.8	4	17.4	3	13.0
Physical agents	8	1.6	5	62.5	3	37.5	3	37.5	5	62.5	2	25.0	6	75.0
Pesticides	19	3.9	10	52.6	9	47.4	9	47.4	10	52.6	3	15.8	14	73.7
Drugs	27	5.5	17	63.0	10	37.0	9	33.3	16	59.3	3	11.1	18	66.7
Solvents	22	4.5	19	86.4	3	13.6	9	40.9	13	59.1	4	18.2	13	59.1
Biological agents	9	1.8	6	66.7	2	22.2	2	22.2	5	55.6	2	22.2	2	22.2
Complex fumes	3	0.6	2	66.7	1	33.3	1	33.3	1	33.3	1	33.3	2	66.7
PAHs	2	0.4	1	50.0	1	50.0	1	50.0	1	50.0	-	-	2	100.0
Chemical reagents	3	0.6	2	66.7	1	33.3	2	66.7	1	33.3	1	33.3	2	66.7
Carbon monoxide	2	0.4	-	-	2	100.0	-	-	2	100.0	-	-	1	50.0
Metals	2	0.4	-	-	2	100.0	1	50.0	1	50.0	1	50.0	1	50.0

**Table 4 ijerph-17-00694-t004:** Description of the occupations for which exposure to reproductive risk factors was identified for employed patients (n = 579) seen in interviews at the ARTEMIS Center from 16 February 2016 to 1 June 2018.

Occupations with Exposure to Reproductive Rik Factors	n	%
Officers of the Armed Forces	1	0.3
Noncommissioned Officers of the Armed Forces	1	0.3
Directors General, Senior Executives, and Members of the Executive and Legislative Bodies	1	0.3
Directors of administrative and commercial services	4	1.2
Directors and Executives, Production and Specialized Services	3	0.9
Specialists in technical sciences	10	3.0
Specialists in justice, social sciences, and culture	5	1.5
Health Specialists	22	6.5
Specialists in business administration	3	0.9
Teaching Specialists	9	2.7
Intermediate occupations in science and technology	7	2.1
Intermediate Health Professions	59	17.5
Intermediate occupations, finance and administration	11	3.3
Intermediate occupations in legal, social, and similar services	2	0.6
Office workers	1	0.3
Reception clerks, counter clerks, and the like	5	1.5
Accounting and procurement employees	2	0.6
Other administrative employees	1	0.3
Direct Personal Services Staff	25	7.4
Service and sales workers	19	5.6
Nursing staff	35	10.4
Farmers and skilled commercial agricultural workers	18	5.3
Subsistence farmers, fishers, hunters and gatherers	1	0.3
Skilled building and related trades, except electricians	14	4.1
Skilled Crafts and Printing Trades	4	1.2
Skilled occupations in metallurgy, mechanical engineering, and the like	6	1.8
Electrical and electro-technical professions	1	0.3
Food processing, woodworking, clothing, and other skilled trades in industry and crafts	6	1.8
Machine and plant operators	5	1.5
Assembly workers	1	0.3
Drivers of vehicles and heavy lifting and maneuvering equipment	2	0.6
Household helpers	37	10.9
Agricultural, fishing, and forestry laborers	3	0.9
Laborers in mining, construction, manufacturing, and transportation industries	5	1.5
Food Manufacturing Assistants	1	0.3
Garbage collectors and other unskilled workers	4	1.2

**Table 5 ijerph-17-00694-t005:** Description of sectors of activity for which exposure to reproductive risk factors was identified for employed patients (n = 579) seen in interviews at the ARTEMIS Center from 16 February 2016 to 1 June 2018.

Sectors of Activity with Exposure to Reproductive Risk Factors	n	%
Activities for human health	82	24.3
Activities of households as employers of domestic workers	1	0.3
Administrative and other business support activities	4	1.2
Air transport	1	0.3
Beverage manufacturing	5	1.5
Chemical industry	2	0.6
Construction and buildings	3	0.9
Creative, artistic, and entertainment activities	2	0.6
Crop and animal production, hunting, and related service activities	24	7.1
Education	17	5.0
Electricity, gas, steam, and air conditioning production and distribution	2	0.6
Food industries	5	1.5
Foodservice	12	3.6
Head office activities; management consulting	3	0.9
Insurance	4	1.2
Land transport and transport by pipeline	3	0.9
Legal and accounting activities	2	0.6
Lodging	3	0.9
Manufacture of computer, electronic, and optical products	1	0.3
Manufacture of electrical equipment	3	0.9
Manufacture of machinery and equipment	1	0.3
Manufacture of metal products, except machinery and equipment	2	0.6
Manufacture of other transport equipment	6	1.8
Medical, social, and social housing	25	7.4
Other nonmetallic mineral product manufacturing	2	0.6
Other personal services	10	3.0
Other specialized, scientific, and technical activities	1	0.3
Postal and courier activities	2	0.6
Printing and reproduction of recordings	2	0.6
Public administration and defense; compulsory social security	15	4.4
Real estate activities	2	0.6
Repair and installation of machinery and equipment	1	0.3
Repair of computers and personal and household goods	1	0.3
Retail trade, except automobiles and motorcycles	28	8.3
Sale and repair of motor vehicles and motorcycles	4	1.2
Scientific research and development	4	1.2
Services related to buildings and landscaping	4	1.2
Social work, exclude accommodation	23	6.8
Specialized construction work	11	3.3
Sports, recreational, and leisure activities	2	0.6
Veterinary activities	1	0.3
Warehousing and services auxiliary to transport	1	0.3
Waste collection, treatment, and disposal; recovery	1	0.3
Water transport	1	0.3
Wholesale trade, except of motor vehicles and motorcycles	5	1.5
Woodworking and manufacture of wooden and cork products, except furniture; manufacture of articles of straw and plaiting materials	2	0.6

**Table 6 ijerph-17-00694-t006:** Nonprofessional exposure reported on the day of consultation of patients seen for fertility disorders at the ARTEMIS Center from 16 February 2016 to 2 May 2019 (n = 302).

	Total		Probability				Frequency				Intensity			
			>50%		≤50%		>30%		≤30%		High		Low	
	n	%	n	%	n	%	n	%	n	%	n	%	n	%
**Residential Environment**														
Agricultural area	268	35.3	242	90.3	22	8.2	232	86.6	8	3.0	10	3.7	35	13.1
Industrial area	21	2.8	16	76.2	5	23.8	20	95.2	-	-	1	4.8	5	23.8
Road traffic	36	4.7	31	86.1	3	8.3	33	91.7	-	-	7	19.4	7	19.4
**Lifestyles**														
Diet	29	3.8	9	31.0	6	20.7	17	58.6	7	24.1	1	3.4	16	55.2
Crafts	103	13.6	97	94.2	6	5.8	40	38.8	44	42.7	14	13.6	61	59.2
Heating	100	13.2	64	64.0	34	34.0	58	58.0	39	39.0	7	7.0	82	82.0
Cosmetics	86	11.3	72	83.7	14	16.3	8	9.3	77	89.5	2	2.3	76	88.4
Housing	39	5.1	27	69.2	8	20.5	31	79.5	2	5.1	1	2.6	3	7.7
Leisure activities	32	4.2	26	81.3	6	18.8	5	15.6	21	65.6	3	9.4	17	53.1
Household-cleaning	570	75.1	519	91.1	51	8.9	298	52.3	200	35.1	4	0.7	556	97.5
Home perfumes	574	75.6	521	90.8	52	9.1	286	49.8	280	48.8	5	0.9	557	97.0
Pesticides	605	79.7	548	90.6	57	9.4	166	27.4	408	67.4	35	5.8	521	86.1
Works	319	42.0	233	73.0	84	26.3	112	35.1	134	42.0	80	25.1	163	51.1

**Table 7 ijerph-17-00694-t007:** Preventive measures proposed to patients or spouses seen at the ARTEMIS Center from 16 February 2016 to 2 May 2019 (n = 779 patients).

**In the Professional Environment**	**n**	**%**
Request and analysis of professional product references	105	13.5
Provision of information on protective equipment	225	28.9
Recommendation to contact the occupational health services following the consultation	230	29.5
Recommendation to contact the occupational doctor during a future pregnancy project	262	33.6
**in the Nonprofessional Environment**		
Request and analysis for references of domestic products	39	5.0
Request for a risk assessment of exposure to lead	9	1.2
Prescription for the measurement of blood lead levels	17	2.2
Contacting of the poison control center	3	0.4
Contacting of pharmacovigilance	6	0.8
Proposal to stop smoking	138	13.8
**Preventive Advice Given in the Report**		
Aeration of housing	179	23.0
Good practices for works	457	58.7
Good housekeeping practices	611	78.4
Limitation of the use of home perfumes	534	68.5
Reduction of pesticide use	448	57.5
Good practices during leisure activities	95	12.2
Advice related to the proximity of the homes to agricultural areas	132	16.9
Zika risk prevention	26	3.3
Tips on home heating	95	12.2
Suppression of the burning of green waste	13	1.7
Advice on the risks associated with the use of alternative water sources	14	1.8

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
