# Peer review of "The ARTEMIS Center: An Environmental Health Prevention Platform Dedicated to Reproduction"

_ijerph, 2020, doi:10.3390/ijerph17030694_

Round 1
Reviewer 1 Report
Please see the attached.

Reviewer 2 Report
This is a very nice description of the investigations in clinical practices. It was very interesting to me how this study is performing in ARTEMIS center. The manuscript is very well written and could be interesting for scientists working in the field of public health. I definitely recommend publishing this paper. However, I would like to present some additional information.
Page 7. -Which pesticides ( or at least, split between insecticides, herbicides, etc.)
-Which solvents?-Which metals?-Which PAH?Describe agricultural areas and domestic pesticide usage
Table 4.- Agricultural, forestry, and fisheries professions are very different, they must be split.-Scientific professionals - authors must clarify if they are working in the labs-Military professions - are they working with chemicals?
Table 5. - How old are houses, What is a history of the usage of flame retardants?
General comment. Why authors did not include in the list of chemical emerging global contaminants, at least, brominated flame retardants?
Again, these suggestions are minor and this manuscript could be recommended for publication.
